# Evaluation of a Novel Biomarker Panel for Acute Kidney Injury Following Endovascular Aortic Repair

**DOI:** 10.3390/ijms262211156

**Published:** 2025-11-18

**Authors:** Konrad Zuzda, Paulina Walczak-Wieteska, Paweł Andruszkiewicz, Jolanta Małyszko

**Affiliations:** 1Department of Nephrology, Dialysis and Internal Medicine, Medical University of Warsaw, 02-097 Warsaw, Poland; konrad.zuzda@wum.edu.pl (K.Z.); jolanta.malyszko@wum.edu.pl (J.M.); 2Doctoral School, Medical University of Warsaw, 02-091 Warsaw, Poland; 32nd Department of Anaesthesiology and Intensive Care, Medical University of Warsaw, 02-097 Warsaw, Poland; pawel.andruszkiewicz@wum.edu.pl

**Keywords:** AKI, EVAR, biomarkers, biomarker panel, PENK, penKid, Semaphorin-3A

## Abstract

Acute Kidney Injury (AKI) following endovascular aortic repair (EVAR) is often diagnosed too late using conventional markers, limiting opportunities for timely intervention in this high-risk population. We investigated whether a mechanism-based biomarker panel could provide improved early AKI detection in EVAR patients. This prospective, single-center study enrolled 68 consecutive EVAR patients between April 2022 and June 2024. AKI was diagnosed using KDIGO 2012 criteria. Seven novel biomarkers, including Proenkephalin A 119-159 (penKid), Semaphorin-3A (SEMA-3A), Retinol Binding Protein-4 (RBP-4), Kidney Injury Molecule-1 (KIM-1), Netrin-1, Tissue Inhibitor of Metalloproteinases-2, and Insulin-Like Growth Factor Binding Protein-7, were measured at baseline, immediate postoperative, 24 h, and 48 h time points, and selected based on distinct nephron locations and release mechanisms. AKI occurred in 18 (26.5%) patients. Top-performing individual biomarkers included serum SEMA-3A (AUC 0.88), serum RBP-4 (AUC 0.81), and penKid (AUC 0.76). A three-biomarker panel combining serum penKid, serum SEMA-3A, and urinary KIM-1 achieved robust discriminatory performance (AUC 0.89, 95% CI 0.77–1.00), superior to individual biomarkers. An alternative panel with serum RBP-4 demonstrated comparable performance (AUC 0.81, 95% CI 0.65–0.99). Multi-biomarker panels combining functional, stress, and injury markers demonstrate promising performance for early AKI detection in EVAR patients. External validation in independent, multi-center cohorts is required before clinical implementation.

## 1. Introduction

Acute kidney injury (AKI) is a critical and common complication following endovascular aortic repair (EVAR), substantially increasing the risk of mortality and morbidity [1,2,3]. The incidence of AKI may be as high as 1 in 3 cases following EVAR procedures, particularly in complex cases involving the implantation of branched and fenestrated aortic stent-grafts [2]. The escalation in perioperative AKI risk with increasing procedural complexity is associated with the implantation of stent-grafts extending to the renal arteries, which may precipitate transient reductions in renal perfusion, an effect exacerbated by exposure to iodinated contrast agents. Notable risk factors for AKI in endovascular aortic interventions include the necessity for perioperative blood transfusion, administration of vasopressor agents, and elevated volumes of intravenous contrast media [4]. Identified patient risk factors encompass female sex, the presence of diabetes mellitus, compromised baseline renal function, and the administration of angiotensin II receptor blockers (ARBs) or angiotensin-converting enzyme (ACE) inhibitors [5,6]. Current approaches to diagnosing AKI remain inadequate for early detection. The syndrome is characterized by rapid-onset renal dysfunction with various causes and is traditionally diagnosed using serum creatinine (SCr) and urine output (UOP) according to KDIGO criteria [7]. However, these gold-standard markers have significant limitations that hinder timely intervention. SCr, despite being widely accessible and inexpensive, suffers from poor sensitivity and delayed response to kidney injury. Its concentration can be confounded by muscle mass, hepatic function, and volume status, often rising up to 48 h after initial injury. UOP similarly proves unreliable, influenced by hypovolemia, fasting, and diuretics rather than true nephron function [8]. Most importantly, neither marker effectively differentiates injury location within the nephron or underlying etiology, limiting phenotype-specific management [9].

Over 75 years [10] of use, SCr has not been replaced by newer markers for AKI diagnosis. Due to the latency in AKI diagnosis, efforts have been made to identify novel kidney stress or injury biomarkers. Introducing novel AKI biomarkers into clinical practice is expected to enhance early diagnosis, surpassing the capabilities of traditional AKI criteria. In addition, their use will allow a more accurate determination of the location of kidney injury and its underlying etiology. This advancement can improve AKI phenotyping and subclinical AKI (sAKI) detection [11,12], which refers to renal impairment that has not yet reached the threshold defined by AKI KDIGO criteria [9]. In addition to expediting the diagnosis of nephron injury, a significant advantage of novel AKI biomarkers is their minimal susceptibility to confounding variables such as gender, age, muscle mass, or comorbidities [13].

The diversity of pathophysiological mechanisms underlying AKI, including glomerular dysfunction, tubular damage, and thrombotic microangiopathy, demands a more sophisticated diagnostic approach. Moreover, traditional criteria fail to identify patients with histopathological renal damage who meet sAKI definitions based on novel biomarker elevations. Several studies have demonstrated successful perioperative AKI prediction and early diagnosis with novel biomarkers [14,15].

We hypothesized that a biomarker panel, selected based on distinct nephron locations and release mechanisms, would provide superior early AKI detection compared to traditional criteria in EVAR patients. This study aimed to evaluate the discriminatory performance of a mechanism-based panel for timely AKI diagnosis in the perioperative period.

## 2. Results

The analysis included 68 adult patients with a median age of 70.5 years (IQR: 62.75, 75.25). All patients underwent EVAR procedures performed by an experienced vascular surgery team at a tertiary referral center. The cohort consisted predominantly of branched EVAR (b-EVAR) procedures (*n* = 60, 88.2%), with fewer standard infrarenal EVAR (*n* = 7, 10.3%) and fenestrated EVAR (f-EVAR) (*n* = 1, 1.5%). During the preoperative period to the third postoperative day, AKI was identified in 18 patients (26.5%). The cohort comprised 25 women (36.8%) and 43 men (63.2%). Men represented a higher proportion in the AKI group (77.8%) versus the non-AKI group (58.3%), *p* = 0.144.

The study cohort exhibited a high burden of cardiovascular comorbidities, reflecting the typical risk profile of patients undergoing EVAR. Median Body Mass Index (BMI) was 26.08 kg/m^2^ (24.31, 29.58) for the overall cohort. No significant difference was observed between the AKI group 26.25 kg/m^2^ (24.98, 30.78) and the non-AKI group 26.27 kg/m^2^ (24.31, 29.58), *p* = 0.751. Hypertension was highly prevalent across both groups, 44 patients (91.67%) in the non-AKI group vs. 17 (94.44%) in the AKI group, *p* = 1.0. Diabetes mellitus affected 13 patients (27.08%) in the non-AKI group vs. 5 (27.78%) in the AKI group, *p* = 1.0. Coronary heart disease was present in 14 (29.17%) non-AKI patients, and in AKI 8 (44.44%) subjects, *p* = 0.241, while chronic obstructive pulmonary disease was present in 5 (27.78%), 4 (8.33%), *p* = 0.055, respectively. While individual comorbidities did not reach statistical significance, associations between specific chronic diseases and AKI occurrence were substantial across the cohort.

Medication profiles revealed several noteworthy associations. The use of low-molecular-weight heparins (LMWH) was significantly higher among patients who developed AKI (16.67% vs. 0%, *p* = 0.018), potentially reflecting patients with more severe vascular disease. Similarly, the preoperative use of alpha-blockers was more prevalent in the AKI group (22.22% vs. 4.17%, *p* = 0.043). Although not reaching statistical significance, a trend toward higher AKI occurrence was observed among patients receiving ARBs (33.33% vs. 12.50%, *p* = 0.073), consistent with the known perioperative risk of renin-angiotensin system blockade. Use of acetylsalicylic acid, ACE inhibitors, beta-blockers, and diuretics showed no significant association with AKI occurrence. The median number of medications taken did not differ between groups (*p* = 0.856), suggesting that polypharmacy alone was not a primary AKI risk factor in this cohort.

Baseline renal function assessed with SCr was significantly elevated in the AKI group, median 1.39 mg/dL, compared to the non-AKI group, median 1.01 mg/dL, *p* < 0.001 (Table 1). Among patients without preexisting chronic kidney disease (CKD), AKI incidence was relatively low, two developed AKI, and 11 did not, *p* = 0.488. However, patients with moderate-to-severe CKD showed markedly elevated AKI risk. Stage 3b CKD was present in 5 (27.78%) of AKI patients compared to only 1 (2.08%) of the non-AKI group (*p* = 0.005). All patients with Stage 4 CKD developed AKI, comprising 16.67% of the AKI group (*p* = 0.018). In contrast, early-stage CKD (Stages 1–2) showed no significant association with AKI occurrence (*p* = 0.556 and *p* = 0.174, respectively).

Among the evaluated markers, several demonstrated strong potential for distinguishing between patients with and without AKI, while others showed limited discriminatory capacity (Table 2 and Appendix C, Figure A1). The penKid exhibited a highly significant difference between groups, with a median = 2.97 ng/mL (2.29, 5.62) in the AKI group compared to median = 1.94 ng/mL (1.69, 2.42) in the non-AKI group (*p* = 0.001). Similarly, sSEMA-3A showed excellent discriminative performance, with a significantly higher concentration in the AKI group, 137.35 pg/mL (83.25, 232.86), compared to the non-AKI group, 69.43 pg/mL (51.99, 82.25), *p* < 0.001.

RBP-4 in both serum and urine also demonstrated significant differences between the two groups. sRBP-4 concentrations were significantly elevated in patients with AKI 24.79 ng/mL (19.13, 28.48) compared to those without AKI 17.11 ng/mL (11.92, 19.67), *p* = 0.004. uRBP-4 levels followed a similar trend, with higher concentrations in the AKI group 289.53 ng/mL (113.35, 377.49) versus the non-AKI group 165.47 ng/mL (66.07, 233.61), *p* = 0.028. PENK/ELISA approached significance (*p* = 0.057), suggesting a potential, albeit weaker, association with AKI. While this marker may still hold some value, its discriminative ability appeared less robust than other markers, such as penKid and SEMA-3A.

Markers such as uKIM-1 (*p* = 0.113), uNetrin-1 (*p* = 0.300), uTIMP-2 (*p* = 0.934), and uIGFBP-7 (*p* = 0.157) did not show statistically significant differences between the AKI and non-AKI groups.

### 2.1. Estimating Optimal Cut-Points for Individual AKI Markers

We conducted an analysis to determine the optimal cut-points for various AKI biomarkers, allowing for better discrimination between patients with and without AKI.

Each biomarker demonstrated varying degrees of discriminative power, as reflected by their respective accuracy, sensitivity, specificity, and Area Under the Curve (AUC) metrics (Table 3).

The penKid biomarker emerged as one of the most promising candidates for clinical use, with an optimal cut-point of ≥2.29 ng/mL. It achieved high accuracy (0.73), sensitivity (0.78), and specificity (0.71) alongside a robust AUC = 0.76. sSEMA-3A also demonstrated exceptional utility, with a cut-point of ≥89.10 pg/mL. It exhibited a high accuracy (0.79) and specificity (0.92), with an AUC = 0.88. The performance of sRBP-4 was similarly strong, with a cut-point of ≥23.72 ng/mL, high specificity (0.92), and an AUC = 0.81. uRBP-4 also offered reasonable performance (AUC 0.74).

Markers such as uKIM-1, uNetrin-1, and uIGFBP-7, while possibly applicable, demonstrated more variable performance. uKIM-1 showed high sensitivity (0.94), making it a strong candidate for early detection of AKI, but its low specificity (0.38) suggests that it may produce a high number of false positives. uNetrin-1 demonstrated perfect specificity (1.00), but its low sensitivity (0.35) limited its utility as a standalone marker. uIGFBP-7 showed balanced sensitivity and specificity (around 0.70), making it a moderately valuable marker, but its AUC (0.66) indicates that it may not be as discriminative as other markers.

### 2.2. Estimating the Agreement Between AKI Diagnosis by KDIGO Criteria and Individual AKI Biomarkers

Agreement between AKI diagnosis by KDIGO criteria and individual biomarker performance was assessed using Gwet’s AC1 statistic, which provides stable estimates in imbalanced datasets (Table 4).

PenKid, with a Gwet’s AC1 = 0.52 (*p* < 0.001), demonstrated moderate agreement with the KDIGO criteria, indicating its potential utility as a reliable marker in clinical practice. Its relatively narrow confidence interval (CI 95%: 0.31–0.73) further supported its stability as a diagnostic tool for AKI. sSEMA-3A showed a slightly higher level of agreement (AC1 = 0.59, *p* < 0.001), suggesting that this marker could offer even better alignment with KDIGO-defined AKI. Conversely, PENK/ELISA showed minimal agreement with the KDIGO criteria (AC1 = 0.10, *p* = 0.428), with a confidence interval that spans negative values, suggesting that this biomarker may not be a reliable indicator of AKI when compared to the KDIGO standard. Similarly, uSEMA-3A (AC1 = 0.23, *p* = 0.392) and uNetrin-1 (AC1 = 0.30, *p* = 0.104) exhibit weak agreement, indicating their diagnostic value may be limited or context-specific.

RBP-4, both in plasma (AC1 = 0.52, *p* = 0.003) and urine (AC = 0.41, *p* = 0.022), demonstrated moderate agreement with the KDIGO criteria. The u RBP-4 also showed reasonable agreement, although with slightly less precision. uKIM-1, with an AC1 of 0.40 (*p* = 0.031), showed moderate concordance. In contrast, uTIMP-2 and uIGFBP-7 showed no meaningful agreement with the KDIGO criteria, with Gwet’s AC1 values of −0.07 and −0.32, respectively.

Correlation analysis between AKI biomarkers was performed to examine their interdependencies (Figure 1). Most biomarker pairs demonstrated weak correlations (coefficients near 0). For instance, penKid and PENK/ELISA showed rho = −0.06, indicating a negligible association. Similar weak correlations were observed between penKid and other biomarkers, including SEMA-3A (plasma and urine), uNetrin-1, and uTIMP-2.

Several biomarker pairs exhibited moderate correlations. uRBP-4 and uKIM-1 demonstrated rho = 0.66 (*p* = 0.003), while uKIM-1 and uTIMP-2. showed rho = 0.63 (*p* = 0.008). The correlation between uTIMP-2 and uIGFBP-7 was rho = 0.53 (*p* = 0.104).

### 2.3. Evaluation of Discriminatory Performance Using AKI Biomarker Panels

In clinical practice, relying on a single biomarker is unlikely to provide the optimal balance of sensitivity and specificity necessary for accurately detecting AKI across diverse patient populations. A multi-biomarker approach could be recommended to enhance diagnostic accuracy and improve clinical outcomes. A panel consisting of penKid, sSEMA-3A, and uKIM-1 offered a promising strategy and was chosen to evaluate discriminatory performance. This combination harnessed the strengths of each biomarker: penKid, which provides balanced sensitivity (0.78) and specificity (0.71); uKIM-1, which showed exceptionally high sensitivity (0.94) for detecting early kidney injury despite modest individual discriminative performance (AUC 0.64); and sSEMA-3A, which exhibited high specificity (0.92) and excellent individual performance (AUC 0.88), collectively enhancing the diagnostic precision of the panel to achieve an AUC of 0.89 (referred to as Panel 1). The inclusion of uKIM-1, despite its lower individual AUC, was based on its complementary role within the multi-marker context.

As a variation (Panel 2) of the panel above, sRBP-4 was considered an alternative to uSEMA-3A in the biomarker panel for AKI detection, forming a second panel for evaluation. RBP-4 offered a similar diagnostic value, particularly due to its specificity, which enhanced the overall performance of the panel but lowered the agreement rate.

The PENK/ELISA demonstrated poor agreement with AKI diagnosis based on KDIGO criteria, indicating a potential limitation in its diagnostic utility.

The composition of the biomarker panels was limited by the sample size (*n* = 29). As a result, the effect values of the individual panel components were not further adjusted for potential confounding variables. The fitted generalized linear models assessed the effect of both biomarker panels on AKI occurrence and demonstrated moderate coefficients of determination, with R^2^ Tjur values of 0.438 and 0.263, respectively.

The receiver operating characteristic (ROC) for Panel 1, comprising penKid, sSEMA-3A, and uKIM-1, demonstrated robust discriminative performance with an ROC of 0.89 (95% CI: 0.77–1.00). In comparison, Panel 2, which included penKid, sRBP4, and uKIM-1, exhibited a slightly lower ROC of 0.81 (95% CI: 0.65–0.99). Despite this difference in AUC, the DeLong test indicated no statistically significant difference between the two ROC curves (Z = 0.68, 95% CI: −0.13–0.27, *p* = 0.494), suggesting that both panels perform similarly regarding AKI detection (Figure 2).

## 3. Discussion

The pathophysiology of AKI is widely described as diverse, encompassing numerous mechanisms including glomerular and tubular damage, and thrombotic microangiopathy [16]. The diagnosis of AKI, as determined by traditional biomarkers, does not provide differentiation based on the underlying pathomechanism. Moreover, KDIGO criteria fail to identify all patients who have histopathological evidence of kidney damage [17]. This particular population has been classified into the sAKI group based on variations in levels of novel biomarkers [18]. To identify the most accurate biomarker panel for early AKI detection, this study evaluated selected ones categorized according to their nephron location and underlying release mechanisms. A biomarker panel consisting of penKid, sSEMA-3A, and uKIM-1 demonstrated promise for early AKI detection and was selected to assess discriminatory performance. This combination harnessed the unique strengths of each biomarker: penKid, which provided balanced sensitivity and specificity; uKIM-1, which showed high sensitivity in detecting early kidney injury; and sSEMA-3A, which exhibited high specificity, collectively enhancing the diagnostic precision of the panel (referred to as Panel 1). As an alternative configuration (Panel 2), sRBP-4 was evaluated as a substitute for sSEMA-3A within the multi-biomarker panel. While sRBP-4 offered comparable diagnostic value due to its specificity, its lower agreement rate influenced overall panel performance.

Studies suggested that SEMA-3A has many functions, such as regulating angiogenesis and organogenesis, but it has also been found in adult podocytes and collecting tubules [19]. SEMA-3A is undetectable in the urine of healthy subjects but can be identified within hours of reperfusion-ischemia injury, and its inactivation suppresses this process [20]. This finding suggests that uSEMA-3A may serve as a marker of renal hypoxic injury [21]. In contrast, the higher sensitivity and specificity of serum rather than urine SEMA-3A were confirmed in our patient cohort. We concluded that urine SEMA-3A may have limited or context-specific diagnostic value. This finding supports the utility of this serum biomarker in perioperative patients at high surgical risk for the reperfusion-ischemic AKI phenotype.

Results from serial measurements of penKid, a commercial set of the functional biomarker PENK, indicate its high utility in the perioperative setting. Our findings are consistent with previous studies [22,23,24]. The function of PENK in the kidney is not fully understood, with possible regulation of diuresis and natriuresis [25,26]. As a functional biomarker, PENK is recognized as a superior option compared to other biomarkers due to its association with glomerular filtration and its minimal tubular influence [26]. Consequently, PENK demonstrates a robust correlation with actual GFR as assessed by the gold standard iohexol method [27].

Both panels, which combined multiple biomarkers, demonstrated relatively high ROC values, reaffirming the value of a multi-biomarker approach over-reliance on a single biomarker. This finding supported the notion that combining biomarkers with diverse strengths, such as sensitivity, specificity, and representation of different biological pathways, enabled a more comprehensive and accurate assessment of kidney function and early AKI detection.

The lack of a statistically significant difference between the two panels, despite the slightly higher ROC for Panel 1 (penKid + sSEMA-3A + uKIM-1) compared to Panel 2 (penKid + sRBP-4 + uKIM-1), suggested that both panels offered comparable diagnostic performance. This indicated that clinicians could choose either panel based on factors such as biomarker availability, cost, sample material, or specific patient characteristics without compromising diagnostic accuracy for AKI detection. The high discriminatory power of both panels (with ROC values exceeding 0.80) highlighted their potential utility in clinical settings to identify AKI at an earlier stage, enabling timely interventions such as fluid management or medication adjustments to mitigate progression to more severe kidney injury.

Additionally, the finding that sRBP-4 could serve as an alternative to sSEMA-3A without significantly affecting the panel’s overall performance provided added flexibility. RBP-4 was significantly elevated in the AKI group but had a lower predictive value than penKid and SEMA-3A. This correlation has not been observed in previous studies [28,29]. In contrast, no significant correlation was found in the cohort for uTIMP-2 and uIGFBP-7, which are reported to be released into the urine shortly after injury [30].

While our biomarker panels demonstrated promising discriminative performance, several factors warrant cautious interpretation. The relatively wide confidence intervals (CIs) for both panels (Panel 1: 0.77–1.00; Panel 2: 0.65–0.99) reflect the limited sample size for panel fitting (*n* = 29) and indicate substantial uncertainty in the point estimates. Additionally, biomarkers commonly experience 5–15 point AUC decreases when validated in independent cohorts, as observed with neutrophil gelatinase-associated lipocalin (NGAL) and TIMP-2×IGFBP-7 biomarker in AKI prediction studies [31,32]. Applying these benchmarks to our results suggests Panel 1 may achieve external validation performance in the range of 0.74–0.84, while Panel 2 may validate at 0.68–0.77. Importantly, the lower confidence bounds of our panels (0.77 and 0.65, respectively) likely provide more realistic estimates of external performance than the point estimates.

The problem of developing an early biomarker panel for perioperative AKI is still in the early research phase [33]. Currently, a commercially available panel of two biomarkers, TIMP-2×IGFBP-7, did not show statistically significant efficacy in our and recent studies [14,34,35]. Adler et al. assessed the feasibility of uTIMP-2×uIGFBP-7 to predict early AKI after out-of-hospital cardiac arrest with promising results [30]. In our cohort, both uTIMP-2 and uIGFBP-7 demonstrated no significant differences between AKI and non-AKI groups (*p* = 0.934 and *p* = 0.157, respectively) and showed negative agreement with KDIGO criteria (AC1 = −0.07 and AC1 = −0.32, respectively), leading to their exclusion from the final panel. In the selection of representative biomarkers for the panels, it was posited that the early diagnosis of perioperative AKI should prioritize well-established injury and functional biomarkers. The biomarker panels proposed in this study were chosen based on the distinct mechanisms influencing the secretion of each biomarker, their varying locations within the nephron, their complementary performance characteristics, and their efficacy in detecting AKI. These promising results open the way for further studies on biomarker panels in perioperative AKI.

The baseline comorbidity profile of our cohort reflects the complex background characteristic of EVAR patients, with substantial cardiovascular risk burden. The strong association between advanced CKD (Stages 3b–4) and AKI development (*p* ≤ 0.018) is particularly clinically relevant, as it identifies patients with eGFR < 45 mL/min/1.73 m^2^ as having severely limited renal reserve and heightened vulnerability to perioperative injury. This finding aligns with established literature demonstrating reduced capacity to compensate for nephrotoxic insults in moderate-to-severe CKD [36]. The significant association between LMWH use and AKI occurrence (*p* = 0.018) warrants careful interpretation; increased bleeding risk compromising renal perfusion, or confounding by indication, whereby patients requiring therapeutic anticoagulation had more severe underlying vascular disease. The trend toward elevated AKI risk with preoperative ARB use (*p* = 0.073) was not consistent with data regarding renin-angiotensin system blockade [37]. These baseline risk factors emphasize the importance of risk stratification and support the clinical utility of our biomarker panels for identifying high-risk patients who may benefit from enhanced monitoring and preventive strategies, considering a 4-fold increase in long-term cardiovascular-specific mortality after vascular surgery [38]. The lack of association between early-stage CKD (Stages 1–2) and AKI suggests that preserved renal function (eGFR ≥ 60 mL/min/1.73 m^2^) provides adequate reserve for most EVAR patients, though biomarker-guided early detection remains valuable even in this population.

The predominantly homogeneous procedural composition of our cohort (88.2% b-EVAR) represents both a strength and a limitation that merits discussion. As a strength, this uniformity minimizes confounding by procedure type that would complicate interpretation in mixed EVAR cohorts. B-EVAR patients undergo similarly complex procedures involving thoracoabdominal aneurysm repair with comparable technical demands, visceral vessel manipulation, and hemodynamic stress patterns [39], providing a relatively uniform baseline of high procedural complexity against which biomarker discrimination could be evaluated. Critically, despite this uniform high procedural complexity, AKI developed in only 26.5% of patients, and our biomarker panels successfully discriminated between those who developed AKI and those who did not (Panel 1 AUC 0.89). This discrimination within a procedurally homogeneous group suggests the biomarkers detect actual kidney injury rather than merely reflecting procedural stress, as the latter would be expected to affect all branched EVAR patients similarly.

Our findings should be interpreted within the broader context of perioperative AKI biomarker research. The NephroCheck™ test, which combines TIMP-2 and IGFBP-7, demonstrated AUCs of 0.80–0.82 for AKI prediction in the multi-center SAPPHIRE study [32], while meta-analyses of NGAL in surgical populations have shown pooled AUCs ranging from 0.71 to 0.78 [31,40]. Our Panel 1 (penKid/sSEMA-3A/uKIM-1) achieved comparable and superior discrimination (AUC 0.89), though direct comparison is limited by differences in patient populations, surgical contexts, and methodological approaches.

### 3.1. Study Strengths and Limitations

This study’s strengths include its prospective design, focus on a clinically relevant high-risk EVAR population, and the evaluation of novel multi-biomarker panels using robust statistical methods. However, the findings are limited by the single-center nature of the research and a modest overall sample size (*n* = 68, with 18 AKI events), particularly for the biomarker panel subgroup analyses (*n* = 29). The limited sample size resulted in wide confidence intervals for panel AUCs (CI widths 0.23–0.34), which reflect substantial uncertainty in performance estimates and increase the risk of optimistic bias despite internal validation efforts. Furthermore, the biomarker panel evaluations could not be adjusted for all potential confounding variables due to sample size restrictions. The predominantly b-EVAR composition (88.2%), while providing procedural homogeneity that minimizes confounding by procedure type, limits generalizability to less complex EVAR variants. While postoperative parameters served as intraoperative stress in this study and showed expected associations with AKI, they cannot fully replace direct intraoperative measurements. This limitation is shared with many biomarker validation studies, where granular procedural data are often unavailable [31,32], but represents an important consideration for interpretation. All limitations necessitate a cautious interpretation of the results as hypothesis-generating and underscore the need for external validation in larger, multi-center studies.

### 3.2. Future Research Directions

Future research should prioritize rigorous external validation of these biomarker panels in larger, multi-center surgical cohorts with diverse patient populations and varied clinical practices. Validation studies should assess not only diagnostic accuracy but also clinical impact, including whether early biomarker-guided detection improves patient outcomes and healthcare resource utilization.

## 4. Methods

This single-center, prospective, observational study enrolled patients undergoing EVAR between April 2022 and June 2024 at the Central Teaching Hospital, Medical University of Warsaw, Poland. The study evaluated whether novel biomarkers could serve as early indicators of kidney dysfunction for timely AKI detection in the post-EVAR period.

### 4.1. Patients Selection

Patients were eligible for enrollment if they met the following criteria: (1) age ≥ 18 years; (2) scheduled for elective endovascular aortic repair; (3) able to provide written informed consent; and (4) available for perioperative follow-up through postoperative day 3.

Patients were excluded if they: (1) required emergency EVAR procedures; (2) had end-stage renal disease requiring chronic dialysis; or (3) were unable to provide adequate urine samples for biomarker analysis. Notably, patients with preexisting CKD or baseline renal impairment were not excluded, as these conditions are prevalent in the EVAR population and reflect real-world clinical practice.

### 4.2. Biomarker Selection

We have selected biomarkers based on the nephron’s location sites and the mechanisms behind their release (Table 5). These include indicators of glomerular filtration, such as Proenkephalin A 119-159 (penKid, PENK) and Retinol Binding Protein-4 (RBP-4), as well as markers associated with tissue damage, mostly with tubular, specifically Kidney Injury Molecule-1 (KIM-1), and Netrin-1, and stress-related markers, including Tissue Inhibitor of Metalloproteinase-2 (TIMP-2), Semaphorin-3A (SEMA-3A), and Insulin-like Growth Factor-Binding Protein-7 (IGFBP-7).

Blood and urine samples were taken perioperatively for up to three consecutive days. Serum and urinary samples were collected simultaneously at each time point (preoperative baseline within 24 h before surgery, during EVAR procedure, 24 h, and 48 h post-procedure), enabling direct temporal comparison of biomarker kinetics across compartments. Plasma for ELISA measurement was collected in ethylenediamine tetraacetic acid (EDTA) tubes, centrifuged at 1300× *g* for 15 min, and stored at −80 °C until batch measurement. Blood samples (5 mL) were collected via venipuncture or existing arterial/venous lines into ethylenediamine tetraacetic acid (EDTA) tubes. Urine samples (10–20 mL) were collected from indwelling catheters. Plasma for ELISA measurement was centrifuged at 1300× *g* for 15 min within 30 min of collection and stored at −80 °C until batch measurement. Not all patients had samples available at every time point due to clinical circumstances, early discharge, catheter removal, or inadequate sample volume.

Serum (sRBP-4, sSEMA-3A) and urine biomarkers (uTIMP-2, uIGFBP-7, uNetrin-1, uKIM-1, uRBP-4) were measured with ELISA kits (R&D Systems, Minneapolis, MN, USA), and PENK was measured with a generic ELISA kit. The assays’ sensitivity, range, and intra-assay precision were described in Appendix A (Table A1, Table A2). PenKid was measured immediately with the IB10 sphingotest^®^ assay, designed as a lateral flow test by SphingoTec GmbH, Hennigsdorf, Germany. It utilized specific monoclonal antibodies to detect PENK levels accurately. The lowest detection limit of the immunoassay was 50 pmol/L.

The diagnosis of AKI was established using recognized clinical criteria, reflecting an increase in SCr levels according to the current KDIGO guidelines [7]. Medical data were collected perioperatively, including demographics, history, laboratory tests, and pharmacological treatments. Baseline SCr measurements were unavailable for 2 patients (2.9%). Given the minimal proportion of missing data (<5% threshold), complete case analysis was performed without imputation, which is appropriate according to established guidelines when missingness is minimal and appears random [47]. For analyses involving baseline SCr and CKD staging, 66 patients with complete data were included. All other baseline and outcome variables had complete data across the full cohort (*n* = 68).

The current study established a significance level of α = 0.05, allowing for a 5% chance of a Type I error. We examined the agreement between AKI diagnosis based on KDIGO criteria and the performance of specific AKI biomarkers. Gwet’s Agreement Coefficient (AC)1 statistic was used to measure the level of agreement. Receiver operating characteristic (ROC) curves were constructed to evaluate the discriminative performance of individual biomarkers and multi-biomarker panels. Area under the curve (AUC) values with 95% confidence intervals were calculated using the DeLong method for biomarker panels. DeLong tests were used to compare ROC curves between panels. Due to the limited sample size relative to the number of potential confounding variables, univariate comparisons were performed for individual biomarkers, and multivariable logistic regression was not conducted to avoid model overfitting and ensure statistical stability. Analyses were conducted using R Statistical Software (version 4.3.3; R Core Team, Vienna, Austria) on Windows 11 Pro 64-bit, utilizing publicly available packages (see Appendix B). A data analysis and statistical plan was written after the data were accessed. No imputation methods were used, as the missing data were assumed to be missing at random and the statistical methods employed (Wilcoxon rank sum test, ROC analysis) inherently accommodate incomplete cases.

The study was approved by the Ethics Committee of the Medical University of Warsaw (8/KBL/OIL/2019 and 53/KBL/OIL/2022), and all participants provided their written informed consent following the Declaration of Helsinki.

## 5. Conclusions

Novel biomarker panels incorporating penKid, sSEMA-3A or sRBP-4, and uKIM-1 demonstrated promising performance for early AKI detection post-EVAR in this exploratory study, outperforming individual markers. These panels show potential as tools to complement traditional risk factors and standard diagnostics for identifying high-risk patients who may benefit from enhanced monitoring and timely intervention. External validation in larger, multi-center cohorts is essential to confirm diagnostic accuracy, assess generalizability across diverse populations and clinical settings, and determine whether the promising performance observed in this study translates to improved clinical outcomes.

## Figures and Tables

**Figure 1 ijms-26-11156-f001:**
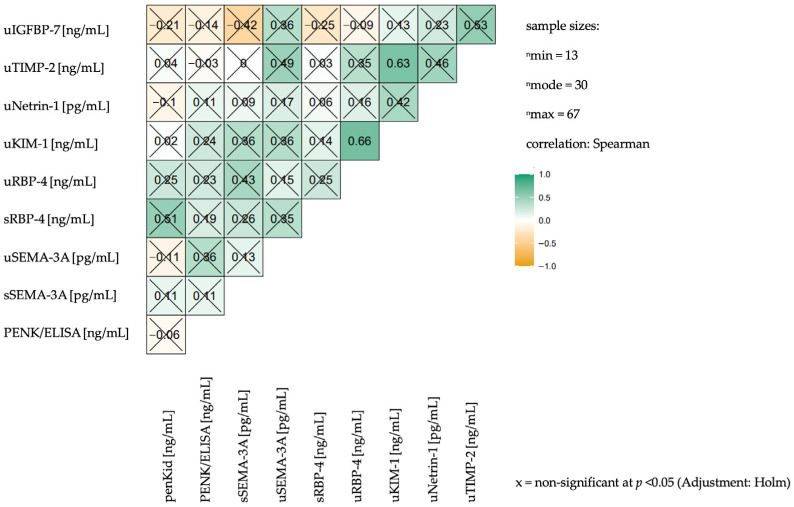
Correlation matrix displaying the relationships between the mean concentrations of AKI markers over the observed period.

**Figure 2 ijms-26-11156-f002:**
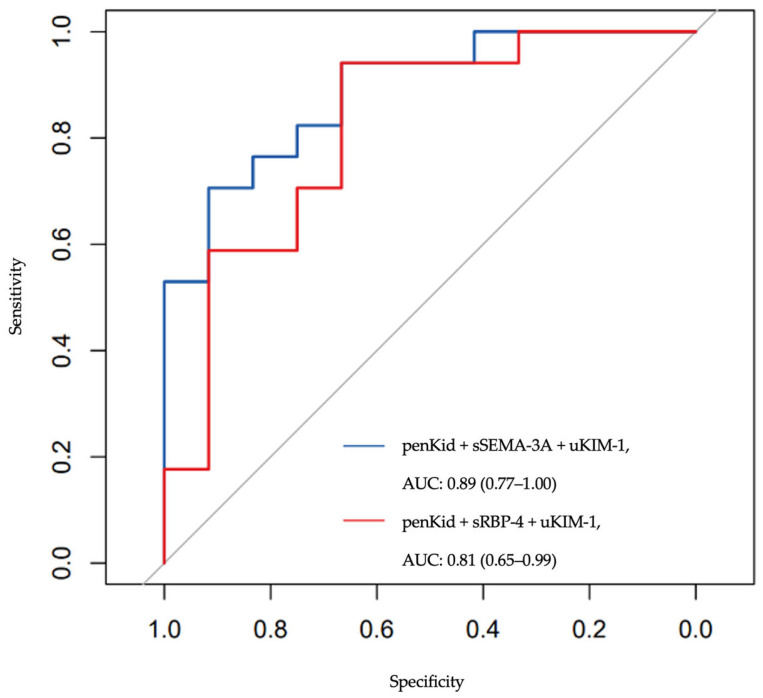
ROC curves for proposed panels of AKI biomarkers.

**Table 1 ijms-26-11156-t001:** Baseline clinical parameters of the study cohort, with stratification by AKI occurrence.

Characteristic	N	Overall	AKI Occurrence		*p*
			Yes, *n*_1_ = 18 ^a^	No, *n*_2_ = 48 ^a^	
SCr ^1^ mg/dl	66 ^f^	1.06 (0.89, 1.21)	1.39 (1.12, 1.83)	1.01 (0.87, 1.15)	<0.001 ^c^
CKD ^2^ stage	66				<0.001 ^c^
no CKD		14 (20.59%) ^b^	2 (11.11%) ^b^	11 (22.92%) ^b^	0.488 ^e^
Stage 1		3 (4.41%) ^b^	0 (0%) ^b^	3 (6.25%) ^b^	0.556 ^e^
Stage 2		31 (45.59%) ^b^	6 (33.33%) ^b^	24 (50%) ^b^	0.174 ^d^
Stage 3a		10 (14.71%) ^b^	1 (5.56%) ^b^	9 (18.75%) ^b^	0.264 ^e^
Stage 3b		6 (8.82%) ^b^	5 (27.78%) ^b^	1 (2.08%) ^b^	0.005 ^e^
Stage 4		3 (4.41%) ^b^	3 (16.67%) ^b^	0 (0%) ^b^	0.018 ^e^
Stage 5		1 (1.47%) ^b^	1 (5.56%) ^b^	0 (0%) ^b^	0.272 ^e^
Body Mass Index	66	26.08 (24.31, 29.58)	26.25 (24.98, 30.78)	26.27 (24.31, 29.58)	0.751 ^c^
Hypertension	66	18 (26.47%) ^b^	17 (94.44%) ^b^	44 (91.67%) ^b^	1.000 ^e^
DM ^3^	66	23 (33.82%) ^b^	5 (27.78%) ^b^	13 (27.08%) ^b^	1.000 ^e^
CHD ^4^	66	11 (16.18%) ^b^	8 (44.44%) ^b^	14 (29.17%) ^b^	0.241 ^e^
COPD ^5^	66	3 (4.41%) ^b^	5 (27.78%) ^b^	4 (8.33%) ^b^	0.055 ^e^
ASA ^6^	66	24 (35.29%) ^b^	6 (33.33%) ^b^	18 (37.50%) ^b^	0.754 ^e^
LMWH ^7^	66	3 (4.41%) ^b^	3 (16.67%) ^b^	0 (0%) ^b^	0.018 ^e^
ACEI ^8^	66	27 (39.71%) ^b^	5 (27.78%) ^b^	22 (45.83%) ^b^	0.184 ^e^
ARB ^9^	66	12 (17.65%) ^b^	6 (33.33%) ^b^	6 (12.50%) ^b^	0.073 ^e^
β-blocker	66	32 (47.06%) ^b^	8 (44.44%) ^b^	24 (50%) ^b^	0.688 ^e^
α-blocker	66	6 (8.82%) ^b^	4 (22.22%) ^b^	2 (4.17%) ^b^	0.043 ^e^
Number of medications	66	3.00 (2.00, 5.00)	3.50 (1.25, 5.00)	3.00 (2.00, 5.00)	0.856 ^d^

^a^ Median (Q1, Q3); ^b^ *n* (%); ^c^ Wilcoxon rank sum test; ^d^ Pearson’s Chi-squared test; ^e^ Fisher’s exact test; ^f^ Two patients from study cohort had not a preoperative SCr measurement; ^1^ SCr—serum creatinine, ^2^ Chronic Kidney Disease, ^3^ Diabetes Mellitus, ^4^ Coronary Heart Disease, ^5^ Chronic Obstructive Pulmonary Disease, ^6^ ASA acetylsalicylic acid, ^7^ LMWH low molecular weight heparin, ^8^ ACEI angiotensin-converting enzyme inhibitors, ^9^ angiotensin II receptor blockers.

**Table 2 ijms-26-11156-t002:** Mean concentrations of AKI biomarkers over time for the overall cohort, stratified by AKI occurrence according to KDIGO criteria.

Characteristic	N	Overall	AKI Occurrence		*p* ^b^
			Yes, *n*_1_ = 18 ^a^	No, *n*_2_ = 48 ^a^	
Baseline—postoperative day 3					
penKid ^1^, ng/mL	67	2.15 (1.73, 2.88)	2.97 (2.29, 5.62)	1.94 (1.69, 2.42)	0.001
PENK/ELISA ^2^ ng/mL	68	1.37 1.06, 2.24)	1.71 (1.24, 2.59)	1.26 (1.01, 1.89)	0.057
Baseline—postoperative day 2					
sSEMA-3A ^3^ pg/mL	29	84.63 (70.40, 178.72)	137.35 (83.25, 232.86)	69.43 (51.99, 82.25)	<0.001
uSEMA-3A ^4^ mL	15	24.32 (19.03, 59.31)	24.60 (18.65, 54.67)	23.88 (23.37, 117.50)	0.594
sRBP-4 ^5^ ng/mL	29	19.85 (16.95, 25.47)	24.79 (19.13, 28.48)	17.11 (11.92, 19.67)	0.004
uRBP-4 ^6^ ng/mL	30	196.11 (97.78, 308.83)	289.53 (113.35, 377.49)	165.47 (66.07, 233.61)	0.028
uKIM-1 ^7^ ng/mL	30	1.98 (1.22, 2.81)	2.20 (1.88, 3.40)	1.69 (0.89, 2.27)	0.113
uNetrin-1 ^8^ pg/mL	30	210.06 (72.64, 519.96)	259.37 (83.46, 728.89)	168.81 (69.03, 230.09)	0.300
uTIMP-2 ^9^ ng/mL	30	6.04 (4.23, 9.08)	4.93 (4.16, 10.00)	6.95 (4.71, 8.88)	0.934
uIGFBP-7 ^10^ ng/mL	30	100.84 (73.13, 151.55)	91.53 (72.33, 139.31)	120.38 (96.64, 172.86)	0.157

^a^ Median (Q1, Q3); ^b^ Wilcoxon rank sum test; ^1^ penKid Proenkephalin A 119-159’s concentration measured with point-of-care IB10 sphingotest^®^ penKid^®^, ^2^ PENK/ELISA Proenkephalin A 159-119’s concentration measured with ELISA assay, ^3^ sSEMA-3A serum Semaphorin-3A, ^4^ uSEMA-3A urinary Semaphorin-3A, ^5^ sRBP-4 serum Retinol Binding Protein-4, ^6^ uRBP-4 urinary Retinol Binding Protein-4, ^7^ uKIM-1 urinary Kidney Injury Molecule-1, ^8^ uNetrin-1 urinary Netrin-1, ^9^ uTIMP-2 urinary Tissue Inhibitor of Metalloproteinases-2, ^10^ uIGFBP-7 urinary Insulin-like Growth Factor-Binding Protein-7.

**Table 3 ijms-26-11156-t003:** Results of optimal cut-point estimations for discriminating patients with AKI with corresponding classification metrics.

AKI Biomarker	*n* _obs_	Optimal Cutoff Point (AKI Occurrence)	CI ^a^ 95%	Accuracy	Sensitivity	Specificity	AUC ^b^
penKid ^1^, ng/mL	67	≥2.29	2.03–3.82	0.73	0.78	0.71	0.76
PENK/ELISA ^2^, ng/mL	68	≥1.16	0.89–2.41	0.56	0.89	0.44	0.65
sSEMA-3A ^3^ pg/mL	29	≥89.10	74.50–179.00	0.79	0.71	0.92	0.88
uSEMA-3A ^4^ pg/mL	15	≥24.31	18.30–∞	0.60	0.60	0.60	0.40
sRBP-4 ^5^ ng/mL	29	≥23.72	13.10–24.80	0.76	0.64	0.92	0.81
uRBP-4 ^6^ ng/mL	30	≥289.53	71.20–377.00	0.70	0.53	0.92	0.74
uKIM-1 ^7^ ng/mL	30	≥1.01	1.01–6.40	0.70	0.94	0.38	0.67
uNetrin-1 ^8^ pg/mL	30	≥706.15	83.50–802.00	0.63	0.35	1.00	0.61
uTIMP-2 ^9^ ng/mL	30	≥4.65	2.37–7.41	0.60	0.47	0.77	0.51
uIGFBP-7 ^10^ ng/mL	30	≥101.70	63.2–257.00	0.70	0.71	0.69	0.66

^a^ CI Confidence Interval, ^b^ AUC Area Under the Curve, ^1^ penKid Proenkephalin A 159-119’s concentration measured with point-of-care IB10 sphingotest^®^ penKid^®^, ^2^ PENK/ELISA Proenkephalin A 159-119’s concentration measured with ELISA assay, ^3^ sSEMA-3A serum Semaphorin-3A, ^4^ uSEMA-3A urinary Semaphorin-3A, ^5^ sRBP-4 serum Retinol Binding Protein-4, ^6^ uRBP-4 urinary Retinol Binding Protein-4, ^7^ uKIM-1 urinary Kidney Injury Molecule-1, ^8^ uNetrin-1 urinary Netrin-1, ^9^ uTIMP-2 urinary Tissue Inhibitor of Metalloproteinases-2, ^10^ uIGFBP-7 urinary Insulin-like Growth Factor-Binding Protein-7.

**Table 4 ijms-26-11156-t004:** The agreement results between AKI diagnosis by KDIGO criteria and individual AKI biomarkers.

AKI Biomarker	Gwet’s AC1 ^a^	SE ^b^	CI ^c^ 95%	*p*
penKid ^1^	0.52	0.11	0.31–0.73	<0.001
PENK/ELISA ^2^	0.10	0.12	−0.15–0.34	0.428
sSEMA-3A ^3^	0.59	0.15	0.28–0.90	<0.001
uSEMA-3A ^4^	0.23	0.26	−0.33–0.79	0.392
sRBP-4 ^5^	0.52	0.16	0.19–0.84	0.003
uRBP-4 ^6^	0.41	0.17	0.06–0.75	0.022
uKIM-1 ^7^	0.40	0.17	0.04–0.76	0.031
uNetrin-1 ^8^	0.30	0.18	−0.07–0.68	0.104
uTIMP-2 ^9^	−0.07	0.20	−0.49–0.34	0.712
uIGFBP-7 ^10^	−0.32	0.18	−0.69–0.031	0.072

^a^ AC1 Agreement Coefficient, ^b^ SE Standard Error, ^c^ CI Confidence Interval, ^1^ penKid Proenkephalin A 159-119’s concentration measured with point-of-care IB10 sphingotest^®^ penKid^®^, ^2^ PENK/ELISA Proenkephalin A 159-119’s concentration measured with ELISA assay, ^3^ sSEMA-3A serum Semaphorin-3A, ^4^ uSEMA-3A urinary Semaphorin-3A, ^5^ sRBP-4 serum Retinol Binding Protein-4, ^6^ uRBP-4 urinary Retinol Binding Protein-4, ^7^ uKIM-1 urinary Kidney Injury Molecule-1, ^8^ uNetrin-1 urinary Netrin-1, ^9^ uTIMP-2 urinary Tissue Inhibitor of Metalloproteinases-2, ^10^ uIGFBP-7 urinary Insulin-like Growth Factor-Binding Protein-7.

**Table 5 ijms-26-11156-t005:** Biomarkers categorized by mechanism of release and nephron location.

Biomarker	Sample Type	Nephron Location	Biomarker Type	Mechanism
RBP-4 ^1^ [28,29]	Urine, Plasma	Proximal Tubule	Injury, Functional	Proximal tubular dysfunction can cause significant increases in urinary RBP-4 due to impaired reabsorption of retinol-free apo-RBP4 fraction.
KIM-1 ^2^ [41]	Urine	Proximal Tubule	Injury	KIM-1 is a transmembrane glycoprotein upregulated in injured proximal tubules. Proteolytic cleavage releases its extracellular domain into urine.
TIMP-2 ^3^ [42]	Urine, Plasma	Distal Tubule	Stress	
IGFBP-7 ^4^ [43]	Urine	Proximal Tubule	Stress	Both IGFBP-7 and TIMP-2 are constitutively expressed in proximal and distal tubules. Urinary elevations result from reduced tubular reabsorption (due to injury) and leakage from damaged cells.
SEMA-3A ^5^ [21,44]	Urine, Plasma	Distal Tubule	Damage	In ischemia–reperfusion-induced AKI, SEMA-3A mediates tissue injury by promoting inflammation and tubular epithelial cell apoptosis. Secreted by injured podocytes and distal tubular cells during AKI.
Netrin-1 [45]	Urine	Proximal Tubule	Damage	Typically, expressed in peritubular capillaries and tubular epithelium. AKI causes downregulation in the vascular endothelium and redistribution to injured tubules.
PENK ^6^ [9,46]	Plasma	Glomerulus	Injury, Functional, Regeneration	PENK accumulates in the plasma in settings of reduced GFR.

^1^ RBP-4 Retinol Binding Protein-4, ^2^ KIM-1 Kidney Injury Molecule-1; ^3^ TIMP-2 Tissue Inhibitor of Metalloproteinases-2, ^4^ IGFBP-7 Insulin-like Growth Factor-Binding Protein-7, ^5^ SEMA-3A Semaphorin-3A, ^6^ PENK/ELISA Proenkephalin A 119-159.

## Data Availability

The complete dataset used in this study has been deposited in the Zenodo repository (https://doi.org/10.5281/zenodo.14827489 (accessed on 5 October 2025)). This dataset is freely available for research purposes under a CC-BY 4.0 license.

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
