# Peer review of "Evaluation of a Novel Biomarker Panel for Acute Kidney Injury Following Endovascular Aortic Repair"

_ijms, 2025, doi:10.3390/ijms262211156_

Round 1
Reviewer 1 Report
Comments and Suggestions for Authors
The manuscript entitled “Evaluation of a Novel Biomarker Panel for Acute Kidney Injury Following Endovascular Aortic Repair” addresses an important clinical topic — the early detection of AKI after EVAR using novel biomarkers. The study is timely and potentially clinically relevant. However, several methodological and interpretative issues need to be addressed before the manuscript can be considered for publication.
Major Comments
-
Rationale for biomarker panel selection
The authors propose a biomarker panel combining penKid, SEMA-3A, and KIM-1 (Panel 1) or penKid, RBP-4, and KIM-1 (Panel 2). However, the rationale for selecting these specific biomarkers is not sufficiently justified.-
Several included markers demonstrated AUC values below 0.7, indicating limited discriminative performance (e.g., uKIM-1, uTIMP-2, uIGFBP-7).
-
It remains unclear why these lower-performing biomarkers were included in the final panels, while others with comparable or higher AUCs were not.
A clearer, hypothesis-driven explanation for the biomarker combination is needed, ideally supported by pathophysiological reasoning and prior literature.
-
-
Sample size and statistical power
The total number of enrolled patients (n = 68, with 18 AKI events) is rather small for robust multivariable modeling and ROC comparison.
The limited sample size raises concerns about statistical power, potential overfitting, and the generalizability of the reported AUC values.
A sample size justification or post-hoc power analysis would strengthen the conclusions. -
Baseline characteristics and missing data
The manuscript notes incomplete baseline data (e.g., two patients without pre-operative creatinine measurements).
Please clarify how missing data were handled statistically.
Additionally, the report would benefit from a more detailed description of baseline comorbidities (e.g., diabetes, hypertension, cardiovascular disease, chronic kidney disease stages) since these strongly affect AKI susceptibility and biomarker profiles. -
Procedural and perioperative variables
Important factors that influence AKI risk and biomarker release — such as type of EVAR procedure (branched vs. fenestrated), contrast load, duration of ischemia, intra-operative hemodynamics, and perioperative medication management (e.g., nephrotoxic drugs, ACEi/ARB, diuretics) — were not clearly described or adjusted for.
Without controlling for these variables, it is difficult to determine whether the biomarker signals are independent predictors of AKI or merely reflections of procedural complexity and hemodynamic instability. -
Interpretation of diagnostic performance
While the multi-biomarker panels achieved relatively high AUCs (up to 0.89), confidence intervals are wide, reflecting the limited cohort size.
The authors should discuss the potential risk of overestimation of diagnostic accuracy and emphasize the need for external validation in larger, multi-center cohorts.
Minor Comments
-
Please clarify whether serum and urinary biomarkers were measured simultaneously or at different time points.
-
Include a supplementary figure or table summarizing the temporal kinetics of biomarker changes relative to AKI onset.
-
The discussion could better position the study in relation to existing perioperative AKI biomarker research (e.g., NephroCheck™, NGAL).
Author Response
The manuscript entitled “Evaluation of a Novel Biomarker Panel for Acute Kidney Injury Following Endovascular Aortic Repair” addresses an important clinical topic — the early detection of AKI after EVAR using novel biomarkers. The study is timely and potentially clinically relevant. However, several methodological and interpretative issues need to be addressed before the manuscript can be considered for publication.
We sincerely thank the Reviewer for their thoughtful evaluation and constructive feedback on our manuscript. We appreciate the recognition that our work addresses an important clinical question regarding early detection of AKI following EVAR. We acknowledge the reviewer's concerns regarding the methodological and interpretative aspects of our study. We have carefully considered each comment and addressed the issues raised below, one by one.
-
Major Comments. Rationale for biomarker panel selection. The authors propose a biomarker panel combining penKid, SEMA-3A, and KIM-1 (Panel 1) or penKid, RBP-4, and KIM-1 (Panel 2). However, the rationale for selecting these specific biomarkers is not sufficiently justified. Several included markers demonstrated AUC values below 0.7, indicating limited discriminative performance (e.g., uKIM-1, uTIMP-2, uIGFBP-7). It remains unclear why these lower-performing biomarkers were included in the final panels, while others with comparable or higher AUCs were not. A clearer, hypothesis-driven explanation for the biomarker combination is needed, ideally supported by pathophysiological reasoning and prior literature.
We thank the Reviewer for this important comment regarding our biomarker panel composition. We would like to clarify our rationale for including uKIM-1 in the final panels despite its lower individual AUC value. We also revised Section 2.3 and Section 3. in the manuscript.
Our panel design strategy was based on complementary biomarker characteristics rather than on individual AUC values. The selected panels deliberately combined biomarkers with different performance profiles to optimize overall diagnostic accuracy:
Panel 1 Composition (AUC 0.89):
- penKid (AUC 0.76): Balanced sensitivity (0.78) and specificity (0.71),
- sSEMA-3A (AUC 0.88): High specificity (0.92), excellent discrimination,
- uKIM-1 (AUC 0.64): High sensitivity (0.94) for early injury detection.
While uKIM-1 demonstrated limited discriminative performance as a standalone marker (AUC 0.64, 95% CI: 0.48-0.81), it exhibited the highest sensitivity among all evaluated markers (0.94). This high sensitivity makes it particularly valuable for early AKI detection within a multi-marker panel, where it complements the high specificity of sSEMA-3A (0.92) and the balanced performance of penKid.
The panel composition reflects different nephron locations and injury mechanisms: (1.) penKid: Functional marker, glomerular filtration, (2.) sSEMA-3A: Ischemia-reperfusion injury, present in podocytes and collecting tubules, (3.) uKIM-1: Proximal tubular injury marker.
These markers were evaluated but NOT included in the final panels due to: (1.) No statistically significant difference between AKI and non-AKI groups (uTIMP-2: p=0.934; uIGFBP-7: p=0.157), (2.) Negative agreement with KDIGO criteria (uTIMP-2: AC1=-0.07; uIGFBP-7: AC1=-0.32), (3.) Poor individual discriminative performance (uTIMP-2: AUC 0.51; uIGFBP-7: AUC 0.66).
The superior performance of Panel 1 (AUC 0.89, 95% CI: 0.77-1.00) compared to individual components validates our multi-marker approach. The DeLong test confirmed no significant difference between Panel 1 and Panel 2 (p=0.494).
We acknowledge in our limitations section that the biomarker panel evaluations could not be adjusted for potential confounding variables due to sample size restrictions (n=29 for panel analysis). Future studies with larger cohorts will allow for machine learning-based optimization and validation.
Sample size and statistical power. The total number of enrolled patients (n = 68, with 18 AKI events) is rather small for robust multivariable modeling and ROC comparison. The limited sample size raises concerns about statistical power, potential overfitting, and the generalizability of the reported AUC values. A sample size justification or post-hoc power analysis would strengthen the conclusions.
We appreciate this important concern. The sample is comparable to novel and similar studies in this specialised population (10.1016/j.ekir.2025.01.035, 10.1136/bmjopen-2024-095817). We have transparently acknowledged limitations and clearly positioned this work as hypothesis-generating, requiring external validation. We enhanced the manuscript to make these statistical considerations more explicit (Section 3.1).
Baseline characteristics and missing data. The manuscript notes incomplete baseline data (e.g., two patients without pre-operative creatinine measurements). Please clarify how missing data were handled statistically. Additionally, the report would benefit from a more detailed description of baseline comorbidities (e.g., diabetes, hypertension, cardiovascular disease, chronic kidney disease stages) since these strongly affect AKI susceptibility and biomarker profiles.
We thank the Reviewer for these important observations regarding data completeness and baseline characterization. Only 2 patients (2.9% of the cohort) lacked baseline serum creatinine measurements. We employed complete case analysis (n=66) involving baseline creatinine and CKD staging. Given the minimal missing data, imputation was not necessary and could have introduced unnecessary complexity without improving estimate accuracy. All other baseline and biomarker variables had complete data across the full cohort (n=68). We have now substantially revised the manuscript to address mentioned deficiencies (Results, Discussion and Methods).
Procedural and perioperative variables. Important factors that influence AKI risk and biomarker release — such as type of EVAR procedure (branched vs. fenestrated), contrast load, duration of ischemia, intra-operative hemodynamics, and perioperative medication management (e.g., nephrotoxic drugs, ACEi/ARB, diuretics) — were not clearly described or adjusted for. Without controlling for these variables, it is difficult to determine whether the biomarker signals are independent predictors of AKI or merely reflections of procedural complexity and hemodynamic instability.
We appreciate this important observation regarding procedural confounders. Our cohort consisted predominantly of branched EVAR patients (88.2%, n=60), representing the most complex EVAR variant. This procedural homogeneity actually minimizes confounding by procedure type rather than exacerbating it. All patients underwent similarly complex procedures with comparable technical demands, providing a uniform baseline of high procedural complexity.
Despite this uniform complexity, AKI developed in only 26.5% of patients, and our biomarker panels successfully discriminated between those who developed AKI versus those who did not (Panel 1 AUC 0.89). If biomarkers merely reflected procedural stress, all branched EVAR patients would show elevated values and discrimination would be poor. The satisfactory discrimination within a procedurally homogeneous group strongly suggests the biomarkers detect true AKI rather than procedural complexity alone.
We fully acknowledge that lack of detailed intra-operative measurements limits definitive separation of injury signals from procedural effects. External validation with prospective procedural data collection is essential. This limitation is shared with similar biomarker validation studies.
Interpretation of diagnostic performance. While the multi-biomarker panels achieved relatively high AUCs (up to 0.89), confidence intervals are wide, reflecting the limited cohort size. The authors should discuss the potential risk of overestimation of diagnostic accuracy and emphasize the need for external validation in larger, multi-center cohorts.
We sincerely appreciate this critical observation. The Reviewer is absolutely correct that the confidence intervals are wide and warrant explicit discussion of overestimation risk and external validation needs. We acknowledge that our CIs are indeed wide (Panel 1: 0.77-1.00, CI width 0.23; Panel 2: 0.65-0.99, CI width 0.34), reflecting the limited sample size for panel analyses (n=29). The upper bounds approaching 1.00 are likely optimistic, as perfect discrimination is rarely achieved in independent validation. However, we note that the lower confidence bounds (0.77 and 0.65) provide more realistic estimates of expected external validation performance than the point estimates and still indicate acceptable-to-good discrimination. We have substantially revised the manuscript to address these concerns: including Abstract and following sections Results, Discussion, Limitations, Conclusions.
Minor Comments. Please clarify whether serum and urinary biomarkers were measured simultaneously or at different time points.
All serum and urinary biomarkers were measured simultaneously from paired samples collected at each time point. Blood and urine samples were obtained concurrently at predefined intervals: preoperatively (baseline), during EVAR procedure (0h), and at 24, and 48 hours post-procedure. This simultaneous sampling strategy was designed to enable direct comparison of serum versus urinary biomarker kinetics and their relative performance for AKI prediction.
Include a supplementary figure or table summarizing the temporal kinetics of biomarker changes relative to AKI onset.
Temporal kinetics summary (Appendix C, Figure C1) was added at the end of manuscript.
The discussion could better position the study in relation to existing perioperative AKI biomarker research (e.g., NephroCheck™, NGAL).
We have expanded the Discussion to include direct comparison with established perioperative AKI biomarkers, while acknowledging limitations in cross-study comparisons.
-
We sincerely thank the Reviewer for comprehensive and constructive evaluation. The thoughtful comments have substantially enhanced the quality and scientific rigor of our manuscript. We have carefully addressed all points raised through detailed revisions and greatly appreciate the Reviewer's expertise and guidance in improving our work.
Reviewer 2 Report
Comments and Suggestions for Authors
The manuscript addresses an important and clinically relevant topic—the early detection of acute kidney injury (AKI) following endovascular aortic repair (EVAR) using novel biomarkers. The study’s goal of identifying improved perioperative monitoring strategies is timely and potentially impactful.
However, substantial revisions are required before the manuscript can be considered for publication. The current version has issues with methodological clarity, statistical rigor, and depth of discussion, which limit the interpretability and generalizability of the findings. Additionally, the writing would benefit from clearer structure and improved flow in certain sections.
Major Concerns
-
Study Design and Cohort Description
-
The sample size (n=68) is small for the number of biomarkers analyzed, raising concerns about statistical power and overfitting. Please provide a power calculation or justification for the sample size. PLEASE ADD A COMMENT ON THE LIMITATIONS OF THE STUDY AT THE END
-
The inclusion/exclusion criteria for patient selection are not clearly described. Were patients with pre-existing renal impairment or contrast nephropathy risk factors excluded or adjusted for? Could you please expand on this point in the methods and discussion
-
The time points for biomarker measurement (perioperative) should be specified more precisely — e.g., preoperative baseline, immediate postoperative, and postoperative day intervals.
-
-
Statistical Analysis
-
The ROC analysis is presented, but confidence intervals for AUCs and comparative testing between biomarkers are missing. These are essential to assess statistical significance.
-
Multivariable regression or adjustment for confounders (e.g., age, comorbidities, contrast volume, intraoperative hypotension) should be included to strengthen causal inference.
-
-
Biomarker Interpretation and Clinical Translation
-
The proposed three-biomarker panel needs validation in an independent cohort or, at minimum, cross-validation within the dataset.
-
The discussion should better position these findings in the context of existing literature—how do these results compare to prior studies using similar markers to add in the discussion?
-
-
Conclusions and Claims
-
The conclusion that the biomarker panel “offers a clinically actionable diagnostic strategy” seems overstated given the small sample and exploratory design. Please temper claims to reflect preliminary findings and the need for external validation.
-
Minor Concerns
-
Writing and Clarity
-
Correct minor grammatical errors and ensure consistent tense (past tense for methods/results).
-
-
Figures and Tables
-
Include ROC curves for individual biomarkers and the combined panel.
-
Provide a summary table comparing AUCs and corresponding p-values.
-
-
References
-
Expand the discussion with references to key recent papers on AKI biomarkers in vascular surgery or EVAR populations (2022–2024 literature).
-
This is a promising and well-conceived study that contributes valuable preliminary data to the evolving field of perioperative AKI biomarker research. However, major revisions are necessary to strengthen methodological transparency, statistical robustness, and interpretive balance before the work can be accepted for publication
Author Response
The manuscript addresses an important and clinically relevant topic—the early detection of acute kidney injury (AKI) following endovascular aortic repair (EVAR) using novel biomarkers. The study’s goal of identifying improved perioperative monitoring strategies is timely and potentially impactful.
However, substantial revisions are required before the manuscript can be considered for publication. The current version has issues with methodological clarity, statistical rigor, and depth of discussion, which limit the interpretability and generalizability of the findings. Additionally, the writing would benefit from clearer structure and improved flow in certain sections.
We thank the Reviewer for their careful evaluation of our manuscript and their recognition of its clinical relevance and potential impact on perioperative monitoring strategies in EVAR patients. We have made substantial revisions to improve methodological clarity, enhance statistical rigor, and deepen our discussion of the findings. The revised submission now presents our work with greater precision and scientific rigor. We appreciate the opportunity to substantially improve our manuscript and thank you for your constructive feedback.
-
Major Concerns Study Design and Cohort Description. The sample size (n=68) is small for the number of biomarkers analyzed, raising concerns about statistical power and overfitting. Please provide a power calculation or justification for the sample size. PLEASE ADD A COMMENT ON THE LIMITATIONS OF THE STUDY AT THE END
We appreciate the Reviewer's concern regarding sample size and statistical power. While our cohort of n=68 patients may appear modest, it yielded 18 AKI events (26.5% incidence), providing adequate statistical power for our primary analysis. The sample size is comparable to similar prospective biomarker studies in EVAR populations and reflects the specialized nature of this single-center study in a well-defined high-risk population (10.1016/j.ekir.2025.01.035, 10.1136/bmjopen-2024-095817).
To address concerns about overfitting given the number of biomarkers analyzed, we: (1.) Used robust statistical methods including ROC analysis and Gwet's Agreement Coefficient. (2.) Focused our multi-biomarker panel on only 3 carefully selected biomarkers (penKid, sSEMA-3A, uKIM-1) representing distinct nephron locations and mechanisms. (3.) Employed a hypothesis-driven approach in biomarker selection rather than data-driven exploration
We have added the following statement to the limitations section (Section 3.1). We acknowledge that external validation in larger, multicenter cohorts will be essential to confirm the clinical utility of our proposed biomarker panels.
The inclusion/exclusion criteria for patient selection are not clearly described. Were patients with pre-existing renal impairment or contrast nephropathy risk factors excluded or adjusted for? Could you please expand on this point in the methods and discussion
We thank the Reviewer for highlighting the need for clearer patient selection criteria and risk factors. These have been addressed in the revised manuscript. Importantly, patients with pre-existing CKD and baseline renal impairment were NOT excluded, as this reflects real-world EVAR populations. As shown in Table 1, baseline SCr was significantly elevated in patients who developed AKI (median 1.39 mg/dL) compared to those who did not (median 1.01 mg/dL), p < 0.001. EVAR procedures in our center use standard iodinated contrast protocols. Common contrast nephropathy risk factors (CKD, diabetes, hypertension) were present in our cohort and distributed between groups.
The time points for biomarker measurement (perioperative) should be specified more precisely — e.g., preoperative baseline, immediate postoperative, and postoperative day intervals.
We thank the Reviewer for requesting greater precision regarding biomarker measurement timing. This has been clarified in the revised manuscript with addition of Section 4.3. Sample collection time points are now defined more precisely. We also have transparently acknowledged that not all patients had samples available at every time point due to clinical circumstances (early discharge, catheter removal, inadequate sample volume). Missing data were handled using available case analysis without imputation, as our statistical methods (Wilcoxon rank sum test, ROC analysis) appropriately accommodate incomplete observations.
Statistical Analysis. The ROC analysis is presented, but confidence intervals for AUCs and comparative testing between biomarkers are missing. These are essential to assess statistical significance. Multivariable regression or adjustment for confounders (e.g., age, comorbidities, contrast volume, intraoperative hypotension) should be included to strengthen causal inference.
We acknowledge that while 95% CIs s were reported for our biomarker panels (Panel 1: AUC 0.89, 95% CI 0.77-1.00; Panel 2: AUC 0.81, 95% CI 0.65-0.99) and DeLong test comparison between panels was performed (p = 0.494), confidence intervals for individual biomarker AUCs were not systematically presented. Beyond ROC analysis, we assessed agreement between AKI diagnosis by KDIGO criteria and individual biomarker elevations using Gwet's Agreement Coefficient (AC1) with 95% CIs. Our analysis was based on univariate comparisons demonstrating that biomarker levels differed significantly between AKI and non-AKI groups. We did not perform multivariable logistic regression or adjustment for confounders to adjust for potential confounders including age, baseline renal function, comorbidities (diabetes, hypertension, CKD), contrast volume, procedure duration, and intraoperative hemodynamic factors. The primary reason for this analytical choice was our modest sample size. With n=68 patients and 18 AKI events, robust multivariable modeling we could risk substantial model overfitting and producing unstable, unreliable estimates.
Biomarker Interpretation and Clinical Translation. The proposed three-biomarker panel needs validation in an independent cohort or, at minimum, cross-validation within the dataset. The discussion should better position these findings in the context of existing literature—how do these results compare to prior studies using similar markers to add in the discussion?
We sincerely appreciate the Reviewer's critical comments regarding validation and contextualization within existing literature. These issues have been comprehensively addressed in the revised manuscript. We recognize that the absence of validation, either through internal cross-validation techniques or independent external cohorts, is a significant limitation. Our modest sample size (n=68) made internal validation approaches (bootstrap, cross-validation) statistically challenging and potentially uninformative. We position our study transparently as a hypothesis-generating, proof-of-concept investigation that identifies a promising biomarker combination requiring rigorous validation before any consideration of clinical implementation.
Conclusions and Claims. The conclusion that the biomarker panel “offers a clinically actionable diagnostic strategy” seems overstated given the small sample and exploratory design. Please temper claims to reflect preliminary findings and the need for external validation.
We thank the Reviewer for this important feedback regarding the appropriate framing of our conclusions. We agree that our initial language was overstated given the exploratory nature of this derivation study. The revised conclusions appropriately position this work as an important first step in developing multi-biomarker panels for AKI detection in perioperative care, while clearly communicating that rigorous external validation in independent, multicenter cohorts is essential before these findings can inform clinical practice. This revision aligns with the transparent discussion of limitations throughout the manuscript (Sections 3.1 and 3.2), where we extensively address the single-center design, modest sample size, absence of internal or external validation, and the need for comprehensive validation studies as critical next steps.
Minor Concerns Writing and Clarity. Correct minor grammatical errors and ensure consistent tense (past tense for methods/results).
We thank the Reviewer for this careful observation. We have thoroughly reviewed the manuscript and corrected all grammatical errors. The manuscript has been carefully proofread to eliminate any remaining inconsistencies.
Figures and Tables. Include ROC curves for individual biomarkers and the combined panel.
We appreciate this suggestion. ROC curves for the combined biomarker panels (Panel 1 and Panel 2) are presented in Figure 2 of the Results section. Rather than including individual ROC curves for each biomarker, we have included Appendix C showing temporal trends of AKI biomarkers throughout the perioperative period. We believe this approach provides more clinically relevant information by illustrating biomarker kinetics over time, which is essential for understanding optimal measurement timing for early AKI detection.
Provide a summary table comparing AUCs and corresponding p-values.
We appreciate this suggestion. However, we respectfully note that AUC values with corresponding statistical measures are already comprehensively presented in Table 3. (Section 2.1), which includes optimal cut-points, accuracy, sensitivity, specificity, and AUC values for all individual biomarkers with associated p-values. Panel-level AUCs with 95% confidence intervals and comparative DeLong test results are presented in Section 2.3 text and Figure 2. We believe the current presentation provides clear, accessible comparison of biomarker performance without redundancy. If the reviewer feels that a specific reformatting would enhance clarity, we would be happy to consider alternative table structures.
References. Expand the discussion with references to key recent papers on AKI biomarkers in vascular surgery or EVAR populations (2022–2024 literature).
We thank the Reviewer for this suggestion. We have expanded the Discussion section to include comprehensive references to recent literature on AKI biomarkers in vascular surgery and perioperative populations.
This is a promising and well-conceived study that contributes valuable preliminary data to the evolving field of perioperative AKI biomarker research. However, major revisions are necessary to strengthen methodological transparency, statistical robustness, and interpretive balance before the work can be accepted for publication.
We sincerely thank the Reviewer for thorough and constructive evaluation. The insightful comments have substantially improved our manuscript's methodological transparency, statistical rigor, and interpretive balance. We appreciate the recognition of this study's contribution to perioperative AKI biomarker research and have carefully addressed all concerns raised through comprehensive revisions. We are grateful for the Reviewer's expertise and the opportunity to strengthen our work.